# Predicting User's Measurements without Manual Measuring: A Case on Sports Garment Applications

**Jochen Vleugels** [1,*] **, Lore Veelaert** [1] **, Thomas Peeters** [1] **, Toon Huysmans** [2,3] **, Femke Danckaers** [3] **and Stijn Verwulgen** [1]

1   Department of Product Development, Faculty of Design Sciences, University of Antwerp, 2000 Antwerp, Belgium
2   Faculty of Industrial Design Engineering, Delft University, 2628 CD Delft, The Netherlands
3   IMEC—Vision Lab, University of Antwerp, 2610 Antwerp, Belgium
*   Correspondence: jochen.vleugels@uantwerpen.be

**Featured Application: Improving garment selection without manual measuring for made-to-order sports garments.**

**Abstract:** As sports garments are stretchable, different sizing tables are used than for retail clothing. However, customers measuring themselves leads to errors and unsatisfaction, since these customized branded garments cannot be returned. Using fitting sets avoids this, but this is not always feasible, especially in an online retail environment. Therefore, this research aims to use descriptive measures—parameters that do not require manual measuring because they are readily known by heart by almost any customer—to predict users' body measurements, which can, thus, be used by customers to determine the size of their sports garment from a sizing chart. To validate if these input measures are sufficient to predict the correct size, three prediction methods are used and compared with baseline manual measurements. The methods are: (i) clothing size predictions from shape models with descriptive measures as inputs, (ii) clothing size predictions from a regression analysis, and (iii) clothing size predictions from a shape model based on extensive 3D scanned measurements as input. The conclusion is that a regression algorithm with, as input variables, the straightforward demographics of age, gender, stature, and weight is more accurate than the algorithm with the same inputs but with a shape model behind it. Moreover, chest and hip circumferences have an intraclass correlation coefficient rating above 0.9 and are, thus, suited for online retail of stretchable garments, such as cycling clothes. As validated by end-users, the regression predictions are shown to agree with preferred garment sizes of the participants, within the natural variation of personal preferences.

**Keywords:** shape model; regression model; customer-tailored sports garment design

## 1. Introduction

The ideal fit of sports apparel depends on the desired properties, such as aerodynamics, comfort, compression, ventilation, and so on. In cycling, for example, both the comfort and aerodynamics take a significant role in the design of the garment [1,2]. Thus, an outfit should wrap around the person's body as tightly as possible without overstretching, and thus, without inducing too much pressure on the body. In this research, a collaboration with sports garment producer Bioracer has given extensive insight in their design rules and aims within the design and development team as well as sales strategies. This paper will use their methods to construct sizing tables and evaluate the fit for cycling garments, without revealing confidential insights. A simple rule of thumb used in this research is that optimal pressure is approximately between 0 and 20% of isotropic stretch (received as a guide by the garment designers). At this stretch, the garment does not put more than 5.0 N of pressure on the body and is never loose. In practice, however, the pressure should also be

adjusted per area of the body and preference of the end user. As an example, the pressure should be lower on the chest and neck area, since this may hinder breathing. However, on the arms and legs, a higher pressure can be helpful to streamline the cyclist, improve posture, and, thus, minimize drag.

Currently, when buying new outfits of the Bioracer brand assortment, amateur cycling teams request a sample set of clothing sizes so that each customer of the team can fit all sizes [1]. In practice, this is often done by hosting one or multiple fitting moments, during which the entire team comes together for test fits. Then, everyone decides what to buy through a personalized online store page (i.e., personalized for the team), or the team lead collects all sizes and places the group order. Consequently, all team members will receive clothing with the same graphics and of the same type, but in their individual sizes. This method makes sure each customer will receive clothing in the correct size; however, this comes with a logistical expense and is not viable in all circumstances, nor is it useful for individual customers. Given the increase in online shopping and the potential of digital solutions, a new method is needed to keep up with the market [3,4]. In practice, the only commercial variant known to the authors is the method of Bivolino [5], which focuses on blouses using a larger set of input parameters to determine clothing size.

In the case study within this article, the clothing is customized for specific cycling teams, and therefore, it is impossible to return the clothing items to the manufacturer without the cost of remaking them. Unfortunately, customers that are unable to attend the fitting session must often decide based on previous knowledge of their optimal size, accept the risk of buying badly fitting garments, or take inaccurate measurements of their body. For this latter purpose, many garment brands provide a lookup table on their online store [6–8], whereby the subject should measure themselves at a couple of points (e.g., chest, waist, and hip circumference) and then approximate their correct size. However, some customers assume they know their sizes based on confection clothing and tend to not use this table correctly. Moreover, this table often varies per clothing piece, style, and type. Thus, sizes cannot be deduced from regular clothing sizes known from confection garments, or even from other garments from the same brand.

## 1.1. Anthropometrics for Customer-Tailored Design

Traditionally, the collection of anthropometric measures was first done in one dimension (1D) by means of calipers to precisely determine the length or width of certain body parts [9–11]. In addition, a measuring tape allowed to assess the circumferences of, for example, the hips or waist. Such data were gathered in 1D anthropometric databases, such as DINED [12] and DINBelg [13], facilitating the comparison of anthropometric data from different studies.

Although few dimensions result in an easy interpretation of the data, there was also a need for more detailed models with greater accuracy, which offer added value to designers that want to include the fitting requirements of their products [11]. Therefore, new techniques and tools focused on 3D geometry to include more details of the human body. Initially, CT scans were used to capture, for example, the variety of the human scalp or head shapes without the interference of hair [11,14]. Of course, disadvantages, such as the required large sample size, feasibility, privacy, and health, had to be considered. Therefore, 3D body scans have also been used to build 3D anthropometric databases, such as the WEAR (World Engineering Anthropometry Resource) annex CAESAR database, leading to the development of human shape models and their deployment in industrial design [11,15].

### 1.1.1. Wear Database

An aid in the set goal of optimizing the fit for the mass public (mass customization of fit) is the WEAR database (formerly known as the CAESAR database). This database was set up two decades ago and still stands as a standard, as a collaboration between three NATO countries (USA, the Netherland, and Italy) to collect ergonomic data about the

general public for the development of product withing the U.S. Air force, The Society of Automotive Engineers (SAE), and others. In contrast to previous collections and database, Caesar was the first to collect state of the art standardized 3D scans of all the subjects, as well as standardized measurements across continents [11]. This database of 3D scans and measurements contains 4000+ subjects from the Netherlands, Spain, and Northern America [15]. It includes 3D scans in a standardized pose and a list of measurements ranging from the basic demographic parameters (e.g., age, gender, ethnicity, etc.) up to detailed measurements taken by researchers (e.g., weight, chest girth, wrist circumference), and so on. All measurements in the database have been recorded as defined in the ISO 7250 nomenclature [16], as well as in their final report on the project [17].

### 1.1.2. Shape Model

To process the massive set of 3D scanned models available in the Wear database, a mathematical analysis should be made. One option for this is to describe the 3D shapes in a statistical shape model (SSM) [14]. In an SSM, all shapes are aligned, such that any two shapes can be compared point-to-point. Representing shapes as a mesh, then the collection of all possible shapes is described as a high dimensional linear space, endowed with a statistical distribution that indicates the probability density for each shape. In a statistical shape model, computational techniques can be applied for, e.g., retrieving the average model, principal component decomposition, etc. Intuitively and practically, an average shape of the population is at the core, and per point in the model, its spatial variation within the population is stored. Despite a shape model's ability to accurately describe the dataset's shapes, it is difficult for non-specialists to interpret in- and outputs and link models to practical use. The model houses a ton of information, but extracting specific needed measurements is not that straightforward. Algorithms are required to match principal components in the shape model with measurements and parameters known to users [14].

In general, using a shape model to predict a user's body shape is possible by combining multiple anthropometric features as inputs and return the shape that best match these inputs. The shape model does this for all points in the 3D mesh model, and thus, compiles a new mesh representation. Various methods are available, depending on the needs [18].

### 1.2. Regression Models

In general terms, regression analysis searches the best matching relationship between given inputs and a desired output. This relation can, depending on the method, be linear in nature or computationally more complex. In this research, regression will be used to model the relation between straightforward demographics, e.g., stature, weight, age, and gender, and a set of output measurements that are used in pattern gradings to determine a subject's clothing size, e.g., chest circumference and hip circumference. In comparison to the shape models, the dimension of the input and output spaces is small compared to the thousands of data points that are used to represent a shape model. This reduces the size of computational data, and thus, adds the possibility to make the regression nonlinear, even with relatively few computational resources.

### 1.3. Research Objective

The use of shape models has proven its use in past research projects, but has mostly been applied for designs on rigid products [14,19,20], e.g., helmets [21], glasses, EEG-headsets [22], headphones, and masks, rather than for textile fabrics. By contrast, this research aims to make the first attempts to provide a comparable assessment and methodology for stretchable, deforming garments that have a different take on fit [23,24]. This article specifically focuses on the case of cycling garments.

Based on the previously mentioned concerns, there are still great opportunities for the optimization of the process to determine a customer's optimal size without having to take measurements or fit the set of clothing. In order to fill this gap, this research aims to

predict a user's garment sizes, by the means of enquiring commonly known parameters (such as age, gender, weight, and stature) to predict chest and hip circumferences, the relevant parameters for cycling garments in this case study. This is predicted through either mathematical shape models or a simplified statistical regression model. To compare additional input options, a 3D scanning method is added to provide additional input measurements to the shape model prediction, thus hoping to increase accuracy with the downside of needing physical measurements. To make statistical comparison possible, all experiments will be compared to their manually measured circumferences that are considered as ground truth throughout the experiment to make statistical analysis possible. Later, the effect on garment sizes is also considered, but with the added complexity of personal preference and personal bias, limiting the statistical insight.

In addition, an extra study has been performed to improve size predictions for female subjects, using the bra size—another parameter that is mostly known by heart—to provide the model with information on the body proportion.

## 2. Materials and Methods

### 2.1. Existing Shape Model

The Shape model used in this article stems from prior research [25–27]. This Shape model is based on 901 of the most complete subjects (317 male and 484 female) from the Dutch subset of the WEAR/CAESAR database, and is filtered based on correct posture, complete mesh, and metadata completeness. The raw scans are processed by rigging to determine correspondence between reference points on both the scan and a reference model. This corrects for errors in posture, and normalizes all scans [27]. In this standardized pose, all points are analyzed in their positional differences compared to the average reference model. The resulting Shape model is based on linear interpolation on all 102.045 points in the mesh model. The database used includes subjects of all ages (18 to 97), statures, body types, and backgrounds, ranging from muscular subjects to obese samples.

The model describes an almost full body model (torso, head, arms, and legs, but no hands or feet), but where needed, subsections of the data can be used to minimize errors from undesired areas. Thus, despite efforts of improved rigging, oftentimes, arms or legs are completely removed from the model, since these have the largest errors that could stray the analysis and conclusions. The shape model holds all the information required to predict the best matching ($L^2$ metric) size and shape of the body that would fit within the given specs or dimensions/parameters entered in the model. For example, providing a user's stature would provide the most average body shape from the model with exactly that stature [27].

In order to predict a random test subject's shape and parameters, the shape model receives a list of measurements or parameters as input, and then tries to predict the best matching size and shape of the subject to fit within these parameters. For this research, the programmed shape model is separated for female and male subjects in the dataset and is developed to return the predictions for 28 different pre-defined measurements [28] as a CSV list to a plain text file and a 3D model to an .stl file containing the same 102,045 points in a mesh, adjusted to this prediction.

Although there is pure math behind the shape models, the input measures and outcomes are approximations in high-dimensional Hilbert space; to make these useful in practice models are used to transform these spaces into measurement approximations. Details on the algorithms, applications in industrial design and useful methodologies behind shape models are build up for rigid products like EEG headsets [22].

For example, when a subject is predicted based on their stature, gender, and age, the weight of the subject is unknown. In that case, the model takes the average, and thus, the most probable shape out of the range of possible postures. However, models need to be programmed to handle a specific set of input measurements, so no optional inputs are possible in practice.

For this research, two prediction models are defined; one that uses four input parameters, i.e., age, gender, stature, and weight, and one that uses the same four parameters with an additional set of five measurements received from the Styku scan. These five are the most accurate measurements from the Styku scanner [1], which are also present in the shape model and feature the same definition: chest circumference, maximal hip circumference, minimal waist circumference, bicep circumference, and neck circumference. In theory, adding more inputs increases the accuracy of the model. However, consideration needs to be taken to prevent overfitting of the model, and whether measurements are defined in the exact same way as in the model [29]. Many of the measurements in Styku are based on features that are not part of the ISO standard [16]. Each additional parameter should increase the accuracy of the prediction a little, but the added benefit might be slim in comparison to the added effort.

As a result of the large amount of data and resolution in the model, this process of predicting takes around two to eight seconds depending on the processor's abilities. These timings do depend on the requirements and optimization in the model.

### 2.2. Participants

Thirty-seven (*n* = 37) Flemish subjects were randomly recruited to participate in the experiment (of which 26 male and 11 female). Ten of these subjects were deemed professional cyclists (of which six male and four female); the others were of varying levels of fitness. The experiments took place individually over a period of one month and lasted approximately 25 min per subject. The ethical commission reviewed the projects goals and granted approval (17/30/345).

### 2.3. Materials

The following materials are required to conduct the research procedure:

- Dressing room and research room;
- A custom clothing set of the Bodyfit range of Bioracer in all sizes;
- Bioracer sizing chart, see Figure 1;
- Measuring tools: calipers, flexible ruler, and scales;
- Laptops to register all data and scans;
- Styku 3D scanner (Styku, n.d.) based on a Kinect V2 scanner and a turntable);
- Shape model of Section 3.1 (including optimal prediction parameters).

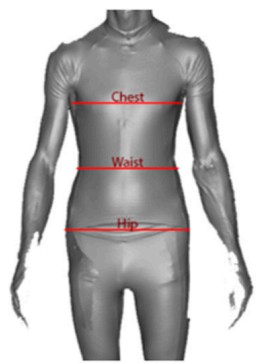

| Pro Bodyfit (men) | Chest (cm) | Waist (cm) | Hip (cm) | INSTRUCTIONS: |
|---|---|---|---|---|
| 0 / XXS | 80 - 85 | 68 - 73 | 80 - 85 | For your chest and hip size: place the measuring tape around the widest part. |
| 1 / XS | 85 - 90 | 73 - 78 | 85 - 90 | |
| 2 / S | 90 - 95 | 78 - 83 | 90 - 95 | For your waist, place the measuring tape just above the belly button around the narrowest spot. |
| 3 / M | 95 - 100 | 83 - 88 | 95 - 100 | |
| 4 / L | 100 - 105 | 88 - 93 | 100 - 105 | |
| 5 / XL | 105 - 110 | 93 - 99 | 105 - 110 | |
| 6 / XXL | 110 - 116 | 99 - 105 | 110 - 116 | If your hips are wider than your chest or vice versa, choose the size that corresponds to the widest of the two. |
| 7 | 116 - 122 | 105 - 111 | 116 - 122 | |
| 8 / SL | 122 - 128 | 111 - 117 | 122 - 128 | |
| 9 | 128 - 134 | 117 - 123 | 128 - 134 | |

**Figure 1.** Example of the sizing chart for male subjects of the professional Bodyfit garment range with instructions on how to measure and select garments (Redrawing from [7]).

### 2.3.1. Clothing Set

The clothing set is based on the regular apparel garments sold by Bioracer with a zipper on the front, but with minimal extra elements, such as pockets, zippers, or seams, see Figure 2. To aid the scanning process, the garment has been printed with a grid pattern

of $1 \times 1$ cm lines. This visually shows stretch in the fabric in both directions that can be measured physically and registered on the scan.

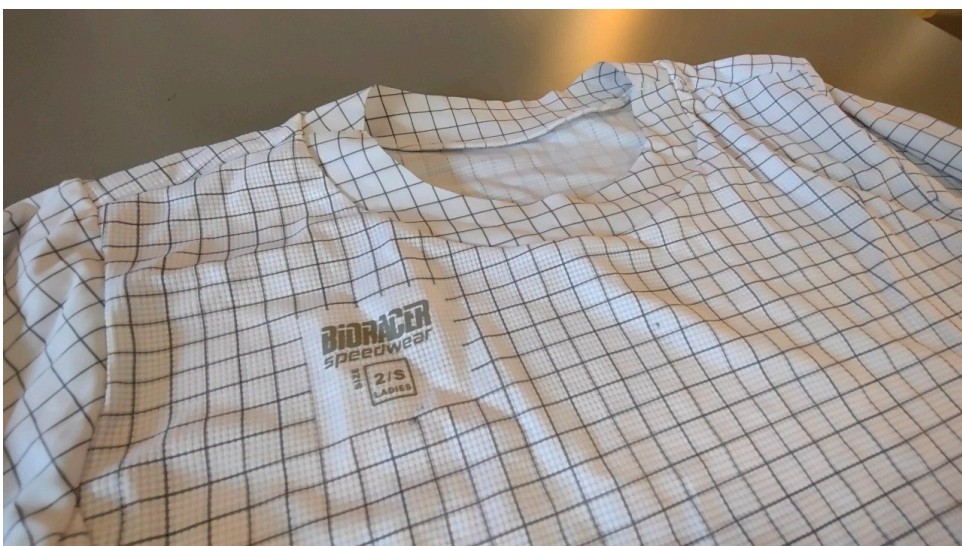

**Figure 2.** Custom made research garments in the standard confection sizes. The custom print consists of a $1 \times 1$ cm grid of lines that aid in 3D scanning and measuring of stretch.

### 2.3.2. Styku 3D Scanner

The Styku scanner is a commercially available scanning system that is mainly oriented on the fitness, health, and wellness sectors [30]. It rotates the subject on a turntable and scans using a Microsoft Kinect V2 scanner, as seen in [31,32]. Its proprietary software requires data input such as body weight and stature, and then scans one full rotation. Data are processed by the proprietary software, and provides a reliable [28] an easy-to-understand overview of common measurements on the body, as well as volume and outline traces [30]. This setup is included as it could be a commercial cost-effective method for receiving relatively accurate measurements with a simplified interface and mode of operation [31,32]. Thus, it can deliver fast (around two minutes) and cheap measurements in a retail or professional environment. However, it has the downside of requiring the subject to be almost naked (underwear) or wearing skin-tight garment (i.e., being almost as time consuming as fitting the real garment in the first place), and a few other restrictions are mentioned in the literature [33–35].

### 2.4. Procedure

First, personal information in the form of four basic parameters was requested from the participants: age, gender, weight, and height.

Second, the chest, waist, and hip circumferences of the participants were roughly measured in normal clothes, in order to select a cycling outfit (pants and shirt) based on the existing size charts from Bioracer (see Figure 1). A set was given to the participant in the supposedly ideal size. Each participant was allowed to try all other outfit sizes in order to determine their optimal fit according to personal preference. In addition, a validation was done on the stretch, using the grid pattern on the outfit. In order to analyze the optimum fitting outfit according to the design rules of Bioracer, it was necessary that as few wrinkles as possible occurred in the worn outfit. Thus, a minimal stretch over the entire outfit was aimed for, while keeping the outfit comfortable for the subject throughout the test. As a consequence, for the further course of the experiment, some participants received a smaller sized clothing to wear in comparison to their supposedly or preferred optimal size.

Third, in addition to the offered outfit, the preferred sizes of the cycling pants and shirts that would have been handpicked by the participants were also noted.

Fourth, based on common measurements that can be found in the WEAR database as well as the shape model, eighteen manual measurements were conducted on the participant wearing the cycling outfits: arm length, shoulder breadth, chest circumference, circumference under bust, hip circumference, waist circumference minimum, waist front length, acromion shoulder breadth, armscye circumference, biceps circumference, chest girth, hip breadth sitting, mid lower arm circumference, neck base circumference, spine to shoulder, tight neck circumference, waist circumference 'at pants height', and wrist circumference. Measurements were done by one and the same trained researcher, to ensure accurate measurements [36]. Anatomical landmarks and measurement methods were followed as described in the CAESAR report [15]. These values were used as reference in this study to compare all predicted measurements with. All measurements were taken in order, twice in a row, and in case of a notable offset in both measurements, a third time to eliminate measuring or typing errors [36]. For the measurements used in the Bioracer sizing table, the closest corresponding Caesar equivalents are used, namely, the maximum hip circumference, the minimal waist circumference, and the chest circumference over bust.

Fifth, wearing the outfit of the second step, two 3D scans with the Styku scanner were taken one after the other. Two scans were used to compare the outcome of both, and, in case of obvious variation in the data, a third scan was taken as validation (this could occur when subjects moved throughout the scanning process). By wearing the cycling jersey, a smooth (wrinkle-free) surface was achieved for the scan, and accurate measurements could be achieved. Based on these 3D scans, 28 measurements of the participant were recorded (for a complete list, see [28,30]).

*2.5. Analysis*

For all comparative results in this research, the manual measurements are taken as a baseline reference to eliminate other influences and ambiguity between methods. Comparing two methods directly with each other would lead to the issue of not knowing what method is the accurate one and which has the offset, since in practice, both will have their own errors. Hence, using the manually measured measurements as reference abstracts the results from the application. Thus, the results can also be compared to other methods and variability in clothing preference is excluded. Therefore, sometimes, extra measurement examples are provided besides only the chest and hip circumference as an additional insight on the accuracy of other measurement predictions. To compare the results, a set of three statistics are selected.

First, the root mean square error (RMSE) is calculated to determine the unsigned difference between the manual measurements and the given prediction model. The RMSE is used to indicate how far off the predictions are in comprehensible units (mm for most measures).

Second, the relative error of all errors is calculated in relation to the base measurement, represented as percentages. With this method, the results are signed, and thus, they provide an indication whether the result is, on average, over- or undersized compared to the original measurements.

Third, the intraclass correlation coefficient (ICC) is determined to give an indication of the predictability of the regression model between the input parameters and the resulting predictions. Overall, a score above 0.75 is considered excellent, and is in this research highlighted in all charts. Values between 0.6 and 0.75 are considered good, but for this research, they are deemed insufficient to use. For use in a real-life commercial application, perhaps even a bar of 0.9 should be reached to fulfil requirements. However, that depends greatly on the circumstances and the allowed margins.

The combination of these three measures gives the best overall insight on the accuracy and precision of the predictions. For example, for the given sports garments in this research, where sizes in the size chart increase in steps of 6 cm, an RMSE of below this 6 cm would be perfectly acceptable. Relative errors could point to consistent deviation in predictions that could, thus, be improved upon by adding extra data samples or offsetting the prediction

results. Finally, the ICC is the most used descriptive statistic method to determine the quality of the prediction.

## 3. Results

This section is structured as follows: in step 1, a regression model is developed based on data of the WEAR/CAESAR database (4094 subjects of the NL, IT, US database). In step 2, two different (extreme) sets of input parameters are selected for the shape model, for the comparative studies of the next step. In step 3, the use of four basic parameters as input for both the manual measurements for the regression model and the shape model is compared to the use of additional detailed data from the Styku scanner in the shape model. To verify the relevance of the simplified regression model, in step 4, both the predicted size according to the regression model and the supposedly ideal size according to the Bioracer size chart are compared to the preferred handpicked size. See Figure 3 for a diagram of the dataflow within the research.

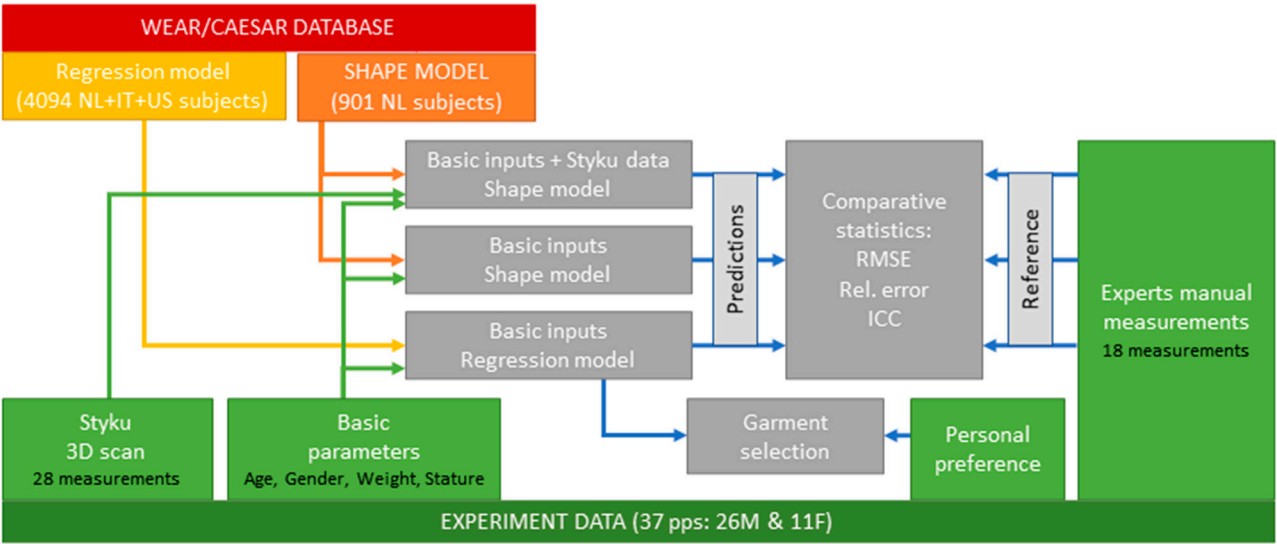

**Figure 3.** Schematic of the dataflow within the research.

### 3.1. Step 1: Regression Model Development

In addition to the existing shape model (see Section 3.1), a non-linear 1D regression-based prediction is developed as well in this research. This implies that 3D scan data would no longer be needed in a commercial setup. This was again built upon the raw data of the WEAR/CAESAR database; however, this time, only the metadata and measurements data were used that were of higher quality and consumed less processing power to compile. Because of missing data, some subjects had to be excluded, but 4194 subjects (from the Netherlands, Spain, and Northern America) of the database remained (2083 male and 2111 female). The shape model needed both the metadata and the 3D scans to be of good quality.

First, the dataset was split into male and female, and per set, the three input parameters (age, stature, and weight) as well as the desired output measurements (as recorded in the WEAR database) were gathered. Of these parameters, log and squared values were calculated. Then, a multiple regression analysis revealed the most optimal predictive formulae to predict the known measurement, see Table 1. The regression model was free to leave out non relevant parameters (i.e., none of the base values, nor the log or squared values were required to be used in the resulting formulae). This analysis gave a list of coefficients to multiply the individual measurements with. Doing so for all of them, and adding them together, provides the most probable prediction within these given input parameters.

**Table 1.** Non-linear regression formulae to predict chest and hip circumferences for male and female subjects.

| Male chest circumference | = | $84.630845 \times \log(age)$ $-0.028169 \times weight^2$ $-0.000204 \times stature^2$ $-4895.536588$ | $+14.480524 \times weigh$ $+2053.400368 \times \log(stature)$ | $-647.094102 \times \log(weight)$ |
|---|---|---|---|---|
| Male hip circumference | = | $0.072907 \times age^2$ $+0.037967 \times weight^2$ $-0000.301 \times stature^2$ $-1311.519809$ | $-10.385996 \times age$ $-8.498566 \times weight$ | $+350.015534 \times \log(age)$ $+1363.200158 \times \log(weight)$ |
| Female chest circumference | = | $0.016543 \times age^2$ $-0.041868 \times weight^2$ $-0000.8065 \times stature^2$ $+1285.248759$ | $-43.189966 \times \log(age)$ $+16.94243 \times weight$ | $-551.488704 \times \log(weight)$ |
| Female hip circumference | = | $-0.339797 \times age$ $+0.025388 \times weight^2$ $-0000.5088 \times stature^2$ $-855.683863$ | $-4.287962 \times weight$ | $+1221.98981 \times \log(weight)$ |

Units: Weight in kg, age in years, stature in mm, and results in mm.

In comparison to the shape model, this regression method focuses on a set number of specific measurements that need to be predicted, rather than the full 100,000 points that are defined in the STL output of the shape model. Reducing the complexity by this much allows less linear models and a more flexible fitting to the complex variation of body sizes and shapes. In the further scope of this research, most focus will be on the garment specific measurements such as chest circumference and hip circumference. However, these regression models can be calculated for any of the measurements that are present in the WEAR database. Nevertheless, the resulting model would need to be analyzed in error to verify whether the input parameters could accurately predict the desired output measurement.

*3.2. Step 2: Selection of Input Parameters for Models*

To select the optimal measures, a prior research project [37] performed on the Wear dataset showed that the most important descriptors are, in order of accuracy of prediction: stature, weight, hip circumference, and gender with an improved accuracy with each added measure. For this research, the aim was to use those descriptive measures that most users know by heart. Thus, the stature, weight, and gender are used together with age. To compare the offset with added measures, an additional analysis was performed on as much input data as could be gathered in a method that could be commercially used. To that end, a Styku scanner was used that provides seven additional measurements on top of the four known features.

In Figure 4, a comparison is made between both shape model prediction methods. The first uses only the base parameters and the second includes five additional measurements from the Styku scanner on top of the base parameters. The graph shows the relative error in comparison to each other as well as the 95% confidence interval. Visible is that the additional Styku data do provide more accurate predictions on most of the measurements, with some exceptions, i.e., the neck circumference and the acronium shoulder breadth. Some improve by a lot with these additional data, since these measurements are the same as those that are predicted by the model. However, issues might come from overfitting, whereby the shape model needs to make weird shapes to fit through the given inputs.

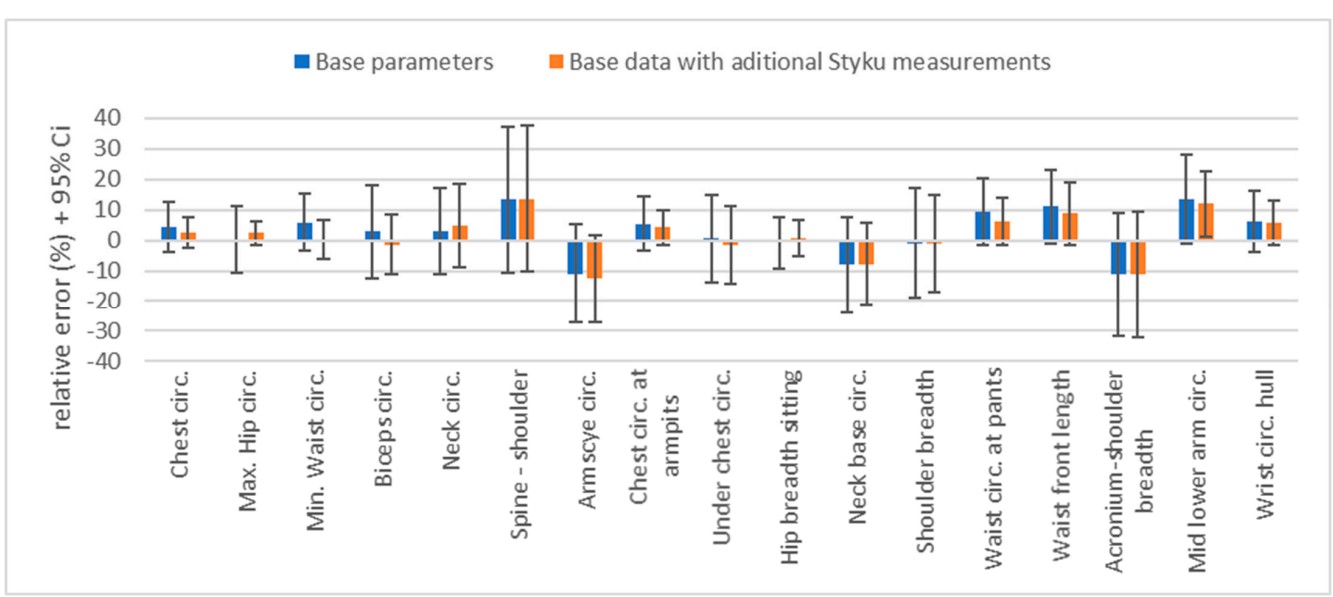

**Figure 4.** Relative error and 95% confidence interval for both shape model prediction methods.

### 3.3. Step 3: Comparison Manual Measurements versus Model Predictions

Using these prediction methods, it would be possible to do a three-way comparison between the various 3D shape model predictions, the non-linear regression model predictions, and the manually taken measurements. However, in order to compare all, the manual measurements were taken as ground truth reference, since they are taken by trained researchers. In the following tables, data are shown as root mean square error, as well as relative error, and the inter correlation coefficient (ICC) shows whether the method is reliable and accurate enough to be effectively used.

Table 2 shows a comparison of the non-linear regression model's results compared to the manually performed measurements on the 37 test subjects. This gave a clear insight that this method was significantly more accurate for male subjects than for female on all measurements, except for the under bust circumference. That one has a slightly better ICC (intraclass correlation coefficient). Based on the ICC for each prediction, only the chest and hip predictions were excellent and would provide sufficiently accurate predictions.

**Table 2.** Overview of results on the regression model prediction based on the base parameter inputs.

| Regression Model (Base Parameters) | RMSE (mm) | | | Relative Error (%) | | | ICC | | |
|---|---|---|---|---|---|---|---|---|---|
| | **M** | **F** | **M+F** | **M** | **F** | **M + F** | **M** | **F** | **M + F** |
| Chest circ. | 24.7 | 49.3 | 34.9 | −0.72 | −2.65 | −1.36 | 0.95 | 0.84 | 0.90 |
| Circ. under Bust | 104.5 | 108.2 | 105.8 | −11.0 | −9.57 | −10.5 | 0.47 | 0.51 | 0.49 |
| Waist circ. | 64.7 | 144.8 | 98.9 | −5.64 | −14.1 | −8.47 | 0.74 | 0.43 | 0.56 |
| Hip circ. | 29.1 | 49.5 | 37.2 | 1.62 | 2.00 | 1.75 | 0.90 | 0.81 | 0.87 |
| Arm length | 127.9 | 86.4 | 115.7 | 24.85 | 16.79 | 22.16 | 0.12 | 0.09 | 0.13 |
| Waist front length | 48.2 | 66.0 | 54.8 | −8.42 | −12.5 | −9.79 | 0.41 | 0.05 | 0.45 |

Table 3 shows the comparison on the shape model predictions using the base parameters (age, gender, weight, and stature). What can be seen is that this model obtains excellent scores on the chest circumference, hip circumference, and circumference under bust, and would, thus, make accurate predictions. However, the difference between male and female

subjects is greater here, showing that for female subjects only, the chest circumference is a usable correlated prediction.

**Table 3.** Overview of the results of the shape model prediction based on the base parameter inputs.

| Shape Model (Base Parameters) | RMSE (mm) | | | Relative Error (%) | | | ICC | | |
|---|---|---|---|---|---|---|---|---|---|
| | **M** | **F** | **M + F** | **M** | **F** | **M + F** | **M** | **F** | **M + F** |
| Chest Circ. | 60.3 | 57.0 | 59.2 | −4.82 | −2.65 | −4.10 | 0.83 | 0.90 | 0.84 |
| Circ. under Bust | 43.2 | 102.2 | 68.7 | 2.91 | −8.44 | −0.87 | 0.86 | 0.57 | 0.76 |
| Waist circ. | 80.1 | 140.9 | 104.4 | −8.04 | −14.12 | −10.07 | 0.72 | 0.50 | 0.60 |
| Hip circ. | 45.9 | 76.1 | 57.7 | −1.74 | 1.60 | −0.63 | 0.82 | 0.54 | 0.75 |
| Arm length | 185.8 | 178.2 | 183.3 | −30.34 | −34.86 | −31.84 | 0.03 | 0.02 | 0.03 |
| Waist front length | 47.0 | 76.0 | 58.3 | −8.65 | −15.22 | −10.84 | 0.48 | 0.12 | 0.48 |

Table 4 shows the comparison on the shape model predictions with the base parameters and additional Styku measurements. This model scores best on a series of measurements: chest circumference, circumference under bust, waist circumference, and hip circumference. It is also capable of predicting all of these with high correlation for both male and female subjects. Only the arm length and waist front length predictions are still very poorly and show no correlation whatsoever.

**Table 4.** Overview of results on the shape model prediction based on the base parameters with additional Styku inputs.

| Shape Model (Base + Styku Data) | RMSE (mm) | | | Relative Error (%) | | | ICC | | |
|---|---|---|---|---|---|---|---|---|---|
| | **M** | **F** | **M + F** | **M** | **F** | **M + F** | **M** | **F** | **M + F** |
| Chest circ. | 33.1 | 39.1 | 35.1 | −2.60 | −2.83 | −2.67 | 0.91 | 0.92 | 0.91 |
| Circ. under Bust | 26.2 | 55.1 | 38.0 | −0.02 | 2.24 | 0.71 | 0.94 | 0.85 | 0.90 |
| Waist circ. | 209.2 | 65.3 | 50.6 | −3.79 | 5.84 | 2.98 | 0.87 | 0.82 | 0.85 |
| Hip circ. | 38.0 | 22.6 | 33.8 | −3.45 | −1.59 | −2.85 | 0.87 | 0.97 | 0.91 |
| Arm length | 280.7 | 242.5 | 269.0 | 56.68 | 47.31 | 53.64 | 0.03 | 0.03 | 0.04 |
| Waist front length | 143.8 | 175.4 | 154.7 | 29.34 | 39.91 | 32.74 | 0.06 | 0.03 | 0.06 |

Figure 5 shows all ICC results combined; the minimal acceptable target is 0.75, although a higher score is better. It should be noted that the root means square error indicated that the shape model with the Styku scanner data was most optimal overall, but not for all measurements. For the waist front length, for some reason, the models based on base data only score much better for males, and thus, also for the combined data. Additionally, it is visible that in a lot of cases, the predictions for females score worse than the ones for males. Depending on the measurement, the results are still accurate, but just less ideal. Thus, considerations need to be taken depending on the target group of the customers population.

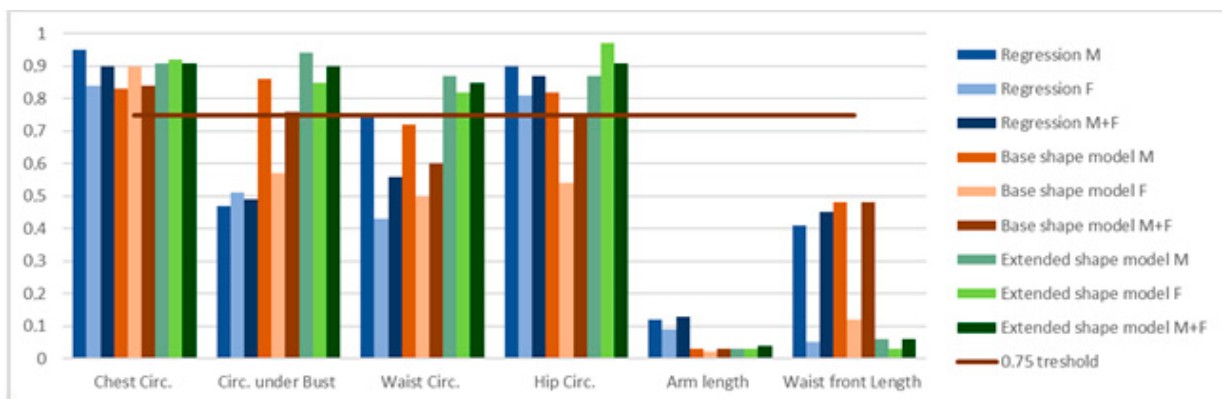

**Figure 5.** ICC values (higher is better) for all three methods in order for all measurements: Regression model (M, F, M + F), Base Shape model (M, F, M + F), and Extended Shape model (M, F, M + F).

### 3.4. Step 4: Comparison Preferred Size versus Model and Chart Predictions

To predict the correct garments based on the predictions from the previous sections, a comparison was performed between the preferred and handpicked clothing sizes, and the ruler measurements and size chart on the one hand, and the regression prediction on the other hand. Both values determined the selection that would be made from the Bioracer size chart [7] based on chest circumference for the shirt and maximal hip circumference for the shorts, as described in their instructions, see Figure 1. These selections were compared to the piece of clothing that the subjects picked themselves, as described above.

Table 5 shows that this resulted in a larger prediction error than the original sizing charts given when compared to the handpicked choices. However, it was visible in the raw data that certain users are way off from the prediction as well as from the measured size. This means that personal preference does play a significant role in the prediction methodology, and should, therefore, be considered in a commercial system. In general, both the measurement-based selection and the predicted selection undersize compared to the preferences of the subjects. This became clear throughout the tests, since plenty of the test subjects stated that they would prefer not to wear such tight clothing. However, the bias of the test could come into play here, since the subjects were explained the goal of the research, and that it should be a sporty fit. Additionally, it a difference between the male and female results was noticeable. Men, on average, chose close to what the prediction and measurements picked, but women were further off on both methods, and thus, seemed to prefer looser shirts but tighter pants. This can also have to do with the prediction model overestimating the hip to chest size in female subjects.

**Table 5.** Comparisons between the handpicked (preferred) garment sizes and the size chart selected ones, as well as between the handpicked and the predicted sizes based on the regression model for both shirts and shorts.

|  | Shirts | | Shorts | |
|---|---|---|---|---|
|  | **Handpicked vs. Size Chart Selection** | **Handpicked vs. Predicted** | **Handpicked vs. Size Chart Selection** | **Handpicked vs. Predicted** |
| **Average error** | −0.11 sizes (0.27 ∣ −1) | −0.43 sizes (−0.08 ∣ −1.27) | −0.19 sizes (−0.69 ∣ 1) | −0.05 sizes (−0.62 ∣ 1.27) |
| **Std** | 0.94 (0.67 ∣ 0.89) | 1.07 (0.84 ∣ 1.1) | 1.05 (0.74 ∣ 0.63) | 1.27 (0.94 ∣ 0.9) |
| **Absolute average** | ±0.65 sizes (±0.42 ∣ ±1.18) | ±0.81 sizes (±0.62 ∣ ±1.27) | ±0.84 sizes (±0.77 ∣ ±1) | ±0.97 sizes (±0.77 ∣ ±1.45) |

Combined results; (Male results ∣ Female results); Positive values are a larger size predicted or measured then chosen.

### 3.5. Female-Specific Adaptations to Regression Model Predictions

Based on this gathered knowledge, a final additional attempt was made to make the regression model more suited for the female subjects. Since chest circumference was debatably the most important measurement for selecting cycling jerseys, the distribution of their body shape and weight depends on the size of the hips and chest and could be inversely proportional to each other when maintaining the same base parameters (age, weight, and stature). Therefore, another measurement should be added to enlighten the prediction model on this balance between shapes. The most known measurement under female subjects could disputably be the bra size (and conveniently, this parameter was also included in the wear database). Crudely, this bra size consists of an under chest circumference and a difference measurement between under and over chest measurement [38]. This measurement would be a perfect fit, since it provides easy insight, as long as the customer is willing to share this personal information to the platform. Bra size charts do change from region to region, standard to standard, and brand to brand, so some effort would be needed to make this a completed solution; however, this analysis is performed as a proof of concept.

In the WEAR database, bra sizes were recorded in EU-size standard. Therefore, the recorded bra size (circumference) was converted from cm to. And the cup size is based on the average conversion values, as defined in the EN 12302 [39], and can be seen in Table 6. Using these results in the formula below in Table 7.

**Table 6.** Cup size to correction dimension between chest circumference and under bust circumference.

| Cup-Size | AA | A | B | C | D | E | F | G | H |
|---|---|---|---|---|---|---|---|---|---|
| Correction Value (mm) | 11 mm | 13 mm | 15 mm | 17 mm | 19 mm | 21 mm | 23 mm | 25 mm | 27 mm |

**Table 7.** Non-linear regression formulae to predict chest and hip circumferences for female subjects, whereby female prediction also uses bra size information.

| | | | |
|---|---|---|---|
| Female chest circumference | = | $0.0007104 \times age^2$ $-0.0015879 \times weight^2$ $-0.0005683 \times stature^2$ $+0.402006 \times bra\_size$ $+22.1669827$ | $+0.7503453 \times weight$ $+0.846651 \times cup\_size\text{-}correction$ |
| Female hip circumference | = | $-0.0272542 \times age$ $+0.0013592 \times weight^2$ $-0.00163 \times stature^2$ $-0.097014 \times bra\_size$ $-452.389437$ | $+88.0784348 \times \log(weight)$ $+138.4747899 \times \log(stature)$ $-0.0113216 \times cup\_size\text{-}correction$ |

Units: Weight in kg, age in years, bra size and cup size in mm, stature in mm, and results in mm.

To validate the improvements, the male and female predictions were compared once more with the new formulae, see Table 8. However, since these data were never requested from the test subjects in the initial study, this analysis cannot be performed on our test subjects' data. The only available sample data are from the wear database, and thus, the same data as used to train the model itself. This was, therefore, tested on a random set of 400 subjects from the Dutch female subset of the wear database and was not comparable to the data in the previous charts that were obtained for real-life subjects. To be able to compare the accuracy to other subjects, the original male regression formulae was also tested again, this time on 400 Dutch male subjects.

**Table 8.** Overview of results on the regression model prediction based on the base parameters with the bra size information.

| Regression Model+ | RMSE (mm) | | | Relative Error (%) | | | ICICC | | |
|---|---|---|---|---|---|---|---|---|---|
| | **M** | **F** | **F + Bra Size** | **M** | **F** | **F + Bra Size** | **M** | **F** | **F + Bra Size** |
| **Chest circumference** | 40.64 | 46.15 | 38.65 | 0.2 | 0.21 | 0.15 | 0.92 | 0.91 | 0.94 |
| **Hip circumference** | 36.42 | 40.61 | 40.23 | 0.12 | 0.15 | 0.17 | 0.91 | 0.92 | 0.93 |

Consequently, Table 8 indicates that the female results now scored better than the male ones, simply because the bra size measurements (assuming the subjects knew their bra size correctly) were a close match to the desired predicted chest circumference measurement. The effects of this intervention might not look like a massive improvement, but it could especially help in the edge cases of the statistics for extreme subjects' situations. However, this must be considered in proportion to the extra personal data being requested of the subjects. This information might be deemed too personal, and not everyone might be willing to provide it; extra care should be taken to inform customers about how the data are used/stored.

## 4. Discussion

Regarding the followed methodology in this paper, several limitations and recommendations for future research should be noted, as well as the relevance to commercial applications and other sectors.

### 4.1. Limitations

First, the shape models used in this study were all based on linear spaces. This was mainly chosen to limit the computational complexity of this model, since non-linear high dimensional models would require many times more data and processing power/time. The shape model in this research was also limited due to the number of subjects in the model, and size constraints to keep the model usable. Since the accuracy of the measurements on both methods should be comparable for a similar subset, the regression method has an advantage in saving processing power, resulting in simple and easy to use measurements based on basic formulae.

Next, within the data that were collected to run the predictions, all common data, such as age, weight, and stature, were based on the input from the subjects themselves, who know these parameters by heart. They were free to weigh or measure themselves, but this was not done per a standard. This way, the prediction was comparable to what customers would enter when using an online system. Notably, subjects disclaimed they might have understated their weight, since they preferred their target weight rather than their actual weight at the time of the experiments. When implementing an online shopping method, this could be a complication that needs to be considered.

### 4.2. Future Opportunities

Regarding future research opportunities, the used grid pattern on the test clothing of the experiment offers possibilities to map the stretch of the clothing in a visual, colored, and graphed way. At the moment, these grid lines on the clothing were added to easily measure the stretch at any given location using a small ruler (i.e., a measurement of 12 mm across two lines would indicate a stretch of 20% on the 10 mm grid). Given the available 3D scan information all around the body, all these data could be combined to analyze stretch in a colormap visualization on the subjects' 3D model. This could aid garment design and customers fitting sessions by visualizing the problem areas, and analyzing the severity of the problem. It can also aid the designers by achieving their desired fit, by showing the

users in a color scale what stretch is desired, and that loose fitting clothes is not advised for optimal performance.

The following two setups can be used to translate the prediction options within this paper to a commercial setup. First, a system to enter personal (base) parameters in a system (application, website, or other). Second, a setup whereby stores would have a Styku scanner system that takes a set of standard measurements on top of those base parameters to improve accuracy. Conversely, in the future, any measurement system (smartphone, webcam, or custom setup) could measure the users body proportions (or even measurements if 3D scanners become implemented in more systems) and add to the model's prediction accuracy.

The advantage in designing for stretch sports garments over confection apparel is the stretchy nature of the fabric. This provides a margin of error whereby the fit is not perfect, but the discomfort is acceptable. When compared to the research of Bivolino [5], this research does not try to modify the garments to fit. Instead, it is aimed to select the optimal garment out of the confection line of garments as shown in the sizing chart of Figure 1. Thus, the degrees of freedom are constrained with less variability in the output. However, the input is also constrained given the limitation of inputs without manual measuring and using known to heart parameters.

For this work, a comparison is made between shape model predictions and regression analysis. An alternative upcoming route is to go for machine learning and thereby allowing additional complexity in the prediction algorithms. However, as is the case with shape models, this also hides the actual dynamic between in- and outputs. The clean relation between the variables is considered a great benefit in real-life applications, since the abilities and limitations are visible.

*4.3. Relevance for Other Sectors*

In comparison to regular apparel that would be bought online, the stretchy characteristics of sports clothing and the fully customized production per client is the key difference. It is impossible to return a clothing item without having to reproduce this item at added cost. For regular online retail, the added benefit of optimizing the selection procedure could possibly save massive amounts of delivery trucks and returned goods. However, some adaptions will be needed, since the measurements used in this experiment might not at all be relevant for normal confection garments.

The proposed methods and analysis would also fit other applications where tight fitting is required, such as pressure clothing used to treat skin burns. These pieces of clothing are made to size using special equipment to avoid having seams on the fabric. At the moment, this is an iterative process to adjust, but it is undesirable to remove the clothing too often in the first moments of treatment, because of the fragile skin during initial treatment. Having a method of predicting the measurements and shape of the person and required patterns could help to reduce manufacturing time and start treatment quicker after the burns without having to perform physical measurements on the patient.

## 5. Conclusions

The fit of cycling apparel depends on desired properties such as comfort and aerodynamics. However, when a fitting moment with all clothing sizes is not feasible or desired, the customer should be able to determine their optimal size without having to fit the specific clothing or use measuring tools and size charts. Therefore, this research project aimed to analyze and compare various solutions to predict a subject's body shape and size for sports garment applications, based on either manual measuring or shape models, in order to validate the use of 3D anthropometric shape models and 1D regression statistics. Advantages of using the shape models are clear, in that the full parameters and details can be used to shape the garment. However, the downsides are also clear, in that the processing of 3D data will always be much more time consuming than analyzing pure 1D measurements through regression analysis.

Therefore, the aim to generate an easy-to-process regression-based model that is accurate enough to predict the users' sports garment is achieved. The regression model with four base parameters, i.e., age, weight, stature, and gender, achieves comparable results on multiple measurements (conveniently on those that are relevant for the sports garment), and is only surpassed by an elaborate shape model method with additional data from a full body 3D scan. The simplicity of not needing to take a physical measurement is a significant advantage, and combined with the fact that the regression models take next to no processing power to analyze, this makes this method the way to go if the data are available and accurate for the needed application.

Especially the need for a 3D scan would be cumbersome, since this already provides all the needed measurements itself. Additionally, the time needed to take the scan could just as well be used to fit the sample garments, which would provide an even better store experience and customer satisfaction.

Specific care should be taken to verify if all needed measurements can accurately be predicted for all populations. For females specifically, for example, extra information is a great advantage in the accurate prediction of their body proportions. For other applications, new parameters might need to be added, such as fitness or body proportion, to further improve predictions.

**Author Contributions:** J.V.: writing—original draft preparation, data curation, writing—review and editing, visualization and validation; L.V.: writing—review and editing and visualization; T.P.: Conceptualization, writing—original draft preparation, methodology, investigation and visualization; T.H.: supervision, resources and data curation; F.D.: resources and software; S.V.: supervision, writing—review and editing, project administration and funding acquisition. All authors have read and agreed to the published version of the manuscript.

**Funding:** This work was supported by IWT-Flanders, Belgium [Virtual size, 2016].

**Institutional Review Board Statement:** This study was approved by the Ethics Committee of UZA/UA (17/30/345, 2017).

**Informed Consent Statement:** Informed consent was obtained from all subjects involved in the study.

**Data Availability Statement:** This study was partially performed using commercially available data from the Wear dataset (formerly known as the Caesar project (1999) [15,17].

**Acknowledgments:** This research was performed with support from Bioracer.

**Conflicts of Interest:** The authors declare no conflict of interest.

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
