# Peer review of "Predicting User’s Measurements without Manual Measuring: A Case on Sports Garment Applications"

_applsci, doi:10.3390/app121910158_

Round 1
Reviewer 1 Report
Although the paper tackles a relevant topic in an useful application, several considerations should be considered:
1. Format of the abstract is not correct.
2. The novelty in the abstract in comparison with previous solutions or measures to tackle the problem should be better highlighted.
3. In the abstract when it is said “decent prediction” in comparison with what or provide a reference for this assessment.
4. In the introduction it is recommended to add more background about similar mathematical challenges and the responses in the literature to validate the gaps. Moreover, it is recommended to specify the gaps more clearly.
5. In the section 2, it is needed a better presentation, specifically in sub-section 2.4 a scheme or figure is needed with steps, input data – methods and output in each step.
6. Steps shown in section 3 should be introduced previously specifying their alignment with the goals of the paper.
7. Section 3 should be improved in its writing and clarity, as for the figures with more clear statements.
8. In section 3, it is needed to clarify the validity of the model taking into account biases and errors always positive or negative.
9. In section 3, tolerance and accuracy influence of devices and methods used during the process.
10. From all the values measures (age, gender, weight and stature) which are the more relevant factors for the prediction in each step of section 3.
11. In section 4 a formal comparison of results with similar topic related researches as well as for similar mathematical challenges should be included.
Best regards
Author Response
Responses in line behind the a. points.
Although the paper tackles a relevant topic in an useful application, several considerations should be considered:
- Format of the abstract is not correct.
- We made it into a single paragraph! Thanks for the remark.
- The novelty in the abstract in comparison with previous solutions or measures to tackle the problem should be better highlighted.
- Added sections on Regression and made changes to try and improve on this. Thanks for the remark.
- In the abstract when it is said “decent prediction” in comparison with what or provide a reference for this assessment.
- Expanded this explication to improve its wording and added the ICC value of >0.9 to specify. Thanks for the remark.
- In the introduction it is recommended to add more background about similar mathematical challenges and the responses in the literature to validate the gaps. Moreover, it is recommended to specify the gaps more clearly.
- Tried to resolve this with an additional section on regression analysis and wording throughout the introduction. Despite not wanting to go to broad in detailing the general possibilities of regression models in general. Thanks for the remark.
- In the section 2, it is needed a better presentation, specifically in sub-section 2.4 a scheme or figure is needed with steps, input data – methods and output in each step.
- A schematic has been made at the start of the research, and has been updated now in the review. But our conclusion is that it did not help in the clarity of the steps. Having short abbreviations of each steps methods and in- and outputs only caused more confusion even within our researchers group. I will continue to try and hopefully come to a clear schematic. But would like each step to stay in long writing as it is now.
- In section 2.5 we did add a step of analysis methods where we explain the processing of the input and outputs of the data.
- Steps shown in section 3 should be introduced previously, specifying their alignment with the goals of the paper.
- In section 2.5 we did add a step of analysis methods where we explain the processing of the input and outputs of the data.
- Section 3 should be improved in its writing and clarity, as for the figures with more clear statements.
- Smaller changes have been made in the figure titles. And parts of the text have been rewritten to hopefully improve on this point. But concrete examples could be helpful if you still see unclear aspects.
- In section 3, it is needed to clarify the validity of the model taking into account biases and errors always positive or negative.
- In section 2.5 an extra paragraph is added to specify the methods and signed/unsigned interpretation of each of the statistics, and how to interpret them.
- In section 3, tolerance and accuracy influence of devices and methods used during the process.
- Manual measurements are taken as ground truth (by a single trained researcher). Parameters are based on subjects memory since these are known to heart (or measurement if they asked). And references about the rulers and 3D scanners accuracy are published about in other research and referenced throughout.
- From all the values measures (age, gender, weight and stature) which are the more relevant factors for the prediction in each step of section 3.
- Added this in section 3.2. It’s also based on previous research from our team (and thus our shape model and dataset) as can be seen in this paper in table one: https://www.researchgate.net/publication/283346959_Evaluation_of_3D_Body_Shape_Predictions_Based_on_Features . For this publication we build on that using the most predictive measurements with the easy to remember ones and compare this to a full stack of measurements.
- In section 4 a formal comparison of results with similar topic related researches as well as for similar mathematical challenges should be included.
- This would be a great subject for a future review article that we have somewhat in the plans. But the options are broad (with Machine learning for example). So for this paper this seemed to much to add for the discussion. But thanks for the remark, definably a future work.
Best regards
Reviewer 2 Report
The submitted manuscript corresponds to the scope of the Special Issue. The paper deals with the urgent problem of measurement of human body, which is timely regarding sustainable development goals.
However, in my opinion, one comment can be added.
1. It will be desired to add Figure and visualize the “chest”, “waist”, and “hip” (please see page 5).
Author Response
Responses in line behind the a. points.
- I recommend authors carefully check hyphenation (eg, line 12 "custom-ized", line 29 "overstretch-ing", and several others).
- To my own dislike I followed the template on this. But in this version the auto hyphenation is disabled and manually implemented where needed. Hopefully the editor clarifies if this is acceptable this way. Thanks for the remark.
- Although the paper states that the Shape model used in this article stems from prior research, I suggest the authors expand the state of the art in terms of techniques: regression models, selection of input parameters for models, etc.
- We added this in section 3.2. The selection of input parameters is also based on previous research from our team (and thus our shape model and dataset) as can be seen in this paper in table one. Personally I believe shape models and their methods behind them are a world of their own. And I prefer to focus on the use cases and let the original maker of the shape models focus on the methods and background how to make them. Thanks for the remark.
- https://www.researchgate.net/publication/283346959_Evaluation_of_3D_Body_Shape_Predictions_Based_on_Features
- It seems to me a good sample of applied research and I have found it pleasant to read. I hope that my comments will be of interest to the authors.
- Thanks!
Reviewer 3 Report
I recommend authors carefully check hyphenation (eg, line 12 "custom-ized", line 29 "overstretch-ing", and several others).
The introduction shows a small state of the art, especially focused on Anthropometrics for Customer-tailored Design. In my opinion, this background refers to the application, but the state of the art on the techniques used is not shown in the paper.
Although the paper states that the Shape model used in this article stems from prior research, I suggest the authors expand the state of the art in terms of techniques: regression models, selection of input parameters for models, etc.
It seems to me a good sample of applied research and I have found it pleasant to read. I hope that my comments will be of interest to the authors.
Author Response
Responses in line behind the a. point:
- It will be desired to add Figure and visualize the “chest”, “waist”, and “hip” (please see page 5).
- Added this to figure 1. Despite the makers of the size chart themselves not being clear about the exact locations. Also added clarity on what measurements we used, based on the Caesar standard measurements.
Thanks for the feedback!
Round 2
Reviewer 1 Report
Although the paper thas been improved, several considerations should be considered before considering for its publication:
2. Wording in the abstract “trough regression”. Please add the novelty specifically.
4. Wording “unput and output”. References need to be added regarding the current possibilities of regression models.
5. In the section 2, it is needed a better presentation, specifically in sub-section 2.4 a scheme or figure is needed with steps, input data – methods and output in each step. It is recommended not to abbreviate the terms but to represent the research design and steps in a figure. With 2.5 you added the content for this purpose.
7. Tables and figures are still to be reworked, for instance table 1 needs to add what 2 means, and how the formulae have been obtained, maybe in an annex for the formulae. Also figure 2 presents overlapping text and boxplots.
8. In section 3, it is needed to clarify the validity of the model taking into account biases and errors yhat need to be explained in the results section
10. From all the values measures (age, gender, weight and stature) which are the more relevant factors for the prediction in each step of section 3. Please specifiy in the novelty and contribution, what are the elements from previous research works and which are the added elements of the research.
11. If a similar topic can be tackled with other mathematical or modelling options and it is not treated in the discussion section. It is recommended to add in the limitations and future research options
Best regards
Author Response
Respones as before in line with the remarks!
- Wording in the abstract “trough regression”. Please add the novelty specifically.
Fixed the spelling error.
We have clarified the narrative throughout the abstract, with the purpose to pinpoint the methods and novelty of our research.
Preliminary to this research, we have constructed shape models from the WEAR data base and observed that some of the parameters that accurately predict the 3D shape of an individual, are actually descriptive of nature and accurately known by heart by most subjects. These parameters are Age, Gender, Stature, and Weight.
Consequently, we constructed a regression model between these descriptive parameters and the measurements that are used to determine a subject’s garment size. It turned out that, with our regression method, the abovementioned descriptive parameters predict the size of cycling garments accurately enough for use in online retail.
- Wording “unput and output”. References need to be added regarding the current possibilities of regression models.
Fixed the spelling error.
We have added references to the mathematical models behind shape models and the approximation algorithms. And our regression model showed the possibility to re-organize online retail in cycling garments since no measurements at all need to be taken.
- In the section 2, it is needed a better presentation, specifically in sub-section 2.4 a scheme or figure is needed with steps, input data – methods and output in each step. It is recommended not to abbreviate the terms but to represent the research design and steps in a figure. With 2.5 you added the content for this purpose.
We agree that the design of experiments could use a clear representation. We have added the request scheme to that end in section 3. Since it fits better in the flow of information there.
- Tables and figures are still to be reworked, for instance table ee1 needs to add what 2 means, and how the formulae have been obtained, maybe in an annex for the formulae. Also figure 2 presents overlapping text and boxplots.
We think that the reveiwer refers to the supercript 2 in the formula, which represents the square of the expression attached lower left. Alternatively, we could opt for a ^2 notation but we think the current notation is more appropriate along the journal's guidelines.
Figure 2 has been reworked so the annotations are lowered.
- In section 3, it is needed to clarify the validity of the model taking into account biases and errors need to be explained in the results section
The manual measurements were taken by an expert and served as ground truth. Further validity is given by ICC statistics. The schema that we added in the section Materials and Methods clarifies this.
- From all the values measures (age, gender, weight and stature) which are the more relevant factors for the prediction in each step of section 3. Please specifiy in the novelty and contribution, what are the elements from previous research works and which are the added elements of the research.
The idea to use descriptive measurements Age, Stature, Weight and Gender is original, yet inspired by previous work where the main factors in a statistical shape models were retreived.
- If a similar topic can be tackled with other mathematical or modelling options and it is not treated in the discussion section. It is recommended to add in the limitations and future research options
Indeed machine learning is likely to play an important role in future “fit without fitting”. We have considered this now in a small paragraph in the discussion section
Best regards